# Differentiable Programming for Earth System Modeling

Maximilian Gelbrecht[1,2], Alistair White[1,2], Sebastian Bathiany[1,2], and Niklas Boers[1,2,3]

[1]Earth System Modelling, School of Engineering & Design, Technical University Munich, Munich, Germany
[2]Potsdam Institute for Climate Impact Research, Potsdam, Germany
[3]Department of Mathematics and Global Systems Institute, University of Exeter, Exeter, UK

**Correspondence:** Maximilian Gelbrecht (maximilian.gelbrecht@tum.de)

**Abstract.** Earth System Models (ESMs) are the primary tools for investigating future Earth system states at time scales from decades to centuries, especially in response to anthropogenic greenhouse gas release. State-of-the-art ESMs can reproduce the observational global mean temperature anomalies of the last 150 years. Nevertheless, ESMs need further improvements, most importantly regarding (i) the large spread in their estimates of climate sensitivity, i.e., the temperature response to increases in atmospheric greenhouse gases, (ii) the modeled spatial patterns of key variables such as temperature and precipitation, (iii) their representation of extreme weather events, and (iv) their representation of multistable Earth system components and the ability to predict associated abrupt transitions. Here, we argue that making ESMs automatically differentiable has huge potential to advance ESMs, especially with respect to these key shortcomings. First, automatic differentiability would allow objective calibration of ESMs, i.e., the selection of optimal values with respect to a cost function for a large number of free parameters, which are currently tuned mostly manually. Second, recent advances in Machine Learning (ML) and in the amount, accuracy, and resolution of observational data promise to be helpful with at least some of the above aspects because ML may be used to incorporate additional information from observations into ESMs. Automatic differentiability is an essential ingredient in the construction of such hybrid models, combining process-based ESMs with ML components. We document recent work showcasing the potential of automatic differentiation for a new generation of substantially improved, data-informed ESMs.

## 1 Introduction

Comprehensive Earth System Models (ESMs) are the key tools to model the dynamics of the Earth system and its climate, and in particular to estimate the impacts of increasing atmospheric greenhouse gas concentrations in the context of anthropogenic climate change (Arias et al., 2021). Despite their remarkable success in reproducing observed characteristics of the Earth's climate system, such as the spatial patterns of the increasing temperatures of the last century, there remain many great challenges for state-of-the-art Earth System Models (Palmer and Stevens, 2019). In particular, further improvements are needed to (i) reduce uncertainties in the models' estimates of climate sensitivity, i.e., temperature increase resulting from increasing atmospheric greenhouse gas concentrations, (ii) better reproduce spatial patterns of key climate variables such as temperature and precipitation, (iii) obtain better representations of extreme weather events, and (iv) be able to better represent multistable Earth system components such as the polar ice sheets, the Atlantic Meridional Overturning Circulation, or the Amazon rainforest, and to reduce uncertainties in the critical forcing thresholds at which abrupt transitions in these subsystems are expected.

With the recent advances in the amount, accuracy, and resolution of observational data, it has been suggested that ESMs could benefit from more direct ways to include observation-based information, e.g. in the parameters of ESMs (Schneider et al., 2017). Systematic and objective techniques to learn from observational data are therefore needed. Differentiable programming, a programming paradigm that enables building parameterized models whose derivative evaluations can be computed via automatic differentiation (AD), can provide such a way of learning from data. AD works by decomposing a function evaluation into a chain of elementary operations whose derivatives are known, so that the desired derivative can be computed using the chain rule of differentiation. Modern AD systems are able to differentiate most typical operations that appear in ESMs, but there also remain limitations. Applying differentiable programming to ESMs means that each component of the ESM needs to be accessible for AD. This enables the possibility to perform gradient- and Hessian-based optimization of ESMs with respect to their parameters, initial conditions, or boundary conditions.

ESMs couple general circulation models (GCMs) of ocean and atmosphere with models of land surface processes, hydrology, ice, vegetation, atmosphere and ocean chemistry and carbon cycle models. To our knowledge, there are currently no comprehensive, fully differentiable ESMs and only select ESM components which are differentiable. Most commonly, these are GCMs used for numerical weather prediction, as they usually need to utilize gradient-based data assimilation methods. These GCMs often achieve differentiability with manually derived adjoint models, and not via AD. However, considerable efforts have also been spent on frameworks such as dolfin-adjoint for finite element models (Mitusch et al., 2019), and the source-to-source AD tools Transforms of Algorithms in Fortran (TAF) (Giering and Kaminski, 1998) and Tapenade (Hascoët and Pascual, 2013) that already enabled differentiable ESM components. A new generation of AD tools such as JAX (Bradbury et al., 2018), Zygote (Innes et al., 2019) and Enzyme (Moses and Churavy, 2020) promise easier use and less user intervention, easier interfacing with Machine Learning (ML) methods and potential co-benefits such as GPU acceleration. Based on these tools, PDE solvers for fluid dynamics have recently been developed (Holl et al., 2020; Kochkov et al., 2021; Bezgin et al., 2023) and advances regarding AD in other fields, such as molecular dynamics (Schoenholz and Cubuk, 2020) and cosmology (Campagne et al., 2023) have been made as well. This is why we review differential programming in this article and outline its potential advantages for ESMs, specifically (but not only) focusing on the integration of ML methods. Differentiable programming seems particularly promising for ESM components that so far often lack adjoint models, so that they are not differentiable. E.g, only few vegetation and carbon cycle models have an adjoint model (Rayner et al., 2005; Kaminski et al., 2013). In this article, we therefore review research that falls in one of three categories: (i) already differentiable GCMs that use AD, (ii) prototypical systems e.g. in fluid dynamics, (iii) differentiable emulators of non-differentiable models, such as artificial neural networks (ANNs) that emulate components of ESMs. We argue that a differentiable ESM could harness most advantages presented in research of all of these categories.

Differentiable programming enables several advantages for the development of ESMs that we will outline in this article. First, differentiable ESMs would allow substantial improvements regarding the systematic calibration of ESMs, i.e., finding optimal values for their $\mathcal{O}(100)$ free parameters, which are currently either left unchanged or tuned manually. Second, analyses of the sensitivity of the model and uncertainties of parameters would benefit greatly from differentiable models. Third, additional information from observations can be integrated into ESMs with ML models. ML has shown enormous potential e.g., for

subgrid parameterization, to attenuate structural deficiencies, and to speed up individual, slow components by emulation. These approaches are greatly facilitated by differentiable models, as we will review in later parts of this article.

Parameter calibration is probably the most obvious benefit of differentiable programming to ESMs. This is why we first review the current state of the calibration of ESMs, before we introduce differentiable programming and automatic differentiation. Thereafter, we will argue for the different benefits that we see for differentiable ESMs and challenges that have to be addressed when developing differentiable ESMs.

## 2   Current State of Earth System Models

Comprehensive ESMs, such as those used for the projections of the Coupled Model Intercomparison Project (CMIP) (Eyring et al., 2016), are highly complex numerical algorithms consisting of hundreds of thousands of lines of Fortran code, which solve the relevant equations (such as the equations of motion) on discrete spatial grids. Mainly associated with processes that operate below the grid resolution, these models have a large number of free parameters that are not calibrated objectively, for example by minimizing a cost function or by applying uncertainty quantification based on a Bayesian framework (Kennedy and O'Hagan, 2001). Instead, values for these parameters are determined via informed guesses and/or an informed trial-and-error strategy often referred to as "tuning". The dynamical core of an ESM relies on fundamental physical laws (conservation of momentum, energy, and mass), and can essentially be constructed without using observations of climate variables. However, uncertain and unresolved processes require parameterizations that rely on observations of certain features of the Earth system, introducing uncertain parameters to the models. This "parameterization tuning" (Hourdin et al., 2017) can be understood as the first step of the tuning procedure. Using short simulations, separate model components such as atmosphere, ocean, vegetation, are typically tuned, after which the full model is fine-tuned by altering selected parameters (Mauritsen et al., 2012). The choice of parameters for tuning is usually based on expert judgment and only a few simulations. The parameters selected for tuning are based on a mechanistic understanding of the model at hand. Suitable parameters have large uncertainty and at the same time exert a large effect on the global energy balance and other key characteristics of the Earth system. Parameters affecting the properties of clouds are therefore among the most common tuning parameters (Mauritsen et al., 2012; Hourdin et al., 2017). Such subjective trial-and-error approaches are common in Earth system modeling because (i) Current ESMs are not designed for systematic calibration, mainly due to limited differentiability of the models. (ii) A sufficiently dense sampling of parameter space by so-called perturbed-physics ensembles with state-of-the-art ESMs is hindered by the unmanageable computational costs. (iii) Varying many parameters makes a new model version less comparable to previous simulations, which is why most parameters are in fact never changed (Mauritsen et al., 2012). (iv) Overfitting would hide compensating errors instead of exposing them, which is needed to improve the models. For example, it is debated how far the modeled 20th century warming is simply the result of tuning, in contrast to being an emergent physical property giving credibility to the models (Mauritsen et al., 2012; Hourdin et al., 2017). The target features that ESMs are usually tuned for are the global mean radiative balance, global mean temperature, some characteristics of the global circulation and sea ice distribution, and a few other large-scale features (Mauritsen et al., 2012; Hourdin et al., 2017). In contrast, regional features of the climate system, and/or features that

are less related to radiative processes, are less constrained by the tuning process. Moreover, current ESMs are typically tuned to reproduce the above target features for recent decades, or the instrumental period of the last 150 years at most. These models therefore have difficulties capturing paleoclimate states (e.g. hothouse or ice age states) or abrupt climate changes evidenced in paleoclimate proxy records (Valdes, 2011). Only for a few examples have modelers recently tuned models successfully to paleoclimate conditions in order to reproduce past abrupt climate transitions (Hopcroft and Valdes, 2021; Vettoretti et al., 2022). The technical challenge associated with tuning complex models not only hinders a more systematic calibration of models and their sub-components, but also makes it difficult to apply them to many scientific questions (e.g., the sensitivity of the climate to forcing) that hinge on a differentiation of the model output. Automatic differentiation therefore suggests itself as a means to making ESMs more tractable.

## 3  Differentiable Programming

Differentiable Programming is a paradigm that enables building parameterized models whose parameters can be optimized using gradient-based optimization (Chizat et al., 2019). The gradients of outputs of such models with respect to their parameters are the key mathematical objects for an efficient parameter optimization. Differentiable Programming allows those gradients to be computed using automatic differentiation (AD). AD was instrumental in the overwhelming success of machine learning methods such as artificial neural networks (ANNs). However, in contrast to pure ANN models, for the wider class of differentiable models one needs to be able to differentiate through control flow and user-defined types. The algorithms used for AD need to exhibit a certain degree of customizability and composability with existing code (Innes et al., 2019). Generally, differentiable programming can incorporate arbitrary algorithmic structures, such as (parts of) process-based models (Baydin et al., 2018; Innes et al., 2019). Several promising projects exist that enable AD of relatively general classes of models. For example, differentiable PDE solvers have been implmenented in Python using the JAX framework (Bradbury et al., 2018; Kochkov et al., 2021). Julia's SciML ecosystem offers differentiable differential equation solvers alongside general-purpose AD systems such as Enzyme.jl or Zygote.jl AD (Rackauckas et al., 2020; Moses and Churavy, 2020; Innes et al., 2019). Specifically for finite element models dolfin-adjoint (Mitusch et al., 2019; Farrell et al., 2013) for the FEniCS (Logg et al., 2012) and Firedrake (Rathgeber et al., 2016) frameworks is available. In order for a model to be differentiable it needs to be written either directly within an appropriate framework (e.g. JAX and dolfin-adjoint), or in a style that conforms to the constraints of the given AD system (e.g. Enzyme.jl or Zygote.jl).

It is important to note that AD is neither a numerical nor a symbolic differentiation: It does not numerically compute derivatives of functions with a finite difference approximation, and does not construct derivatives from analytic expressions like computer algebra systems. Instead, AD computes the derivative of an evaluation of some function of a given model output, based on a non-standard execution of its code so that the function evaluation can be decomposed into an evaluation trace or computational graph that tracks every performed elementary operation. Ultimately, there is only a finite set of elementary operations such as arithmetic operations or trigonometric functions and the derivatives of those elementary operations are known to the AD system. Then, by applying the chain rule of differentiation, the desired derivative can be computed. AD

systems can operate in two different main modes: a forward mode, which traverses the computational graph from the given input of a function to its output, and a reverse mode, which goes from function output to input. Reverse-mode AD achieves better scalability with the input size, which is why it is usually preferred for optimization tasks that usually only have a single

output – a cost function – but many inputs (see e.g. Baydin et al. (2018) for a more extensive introduction to AD). Most modern AD systems directly provide the possibility to compute gradients of user-defined functions with respect to chosen parameters at some input value. They do not require any further user action. However, which functions are differentiable by AD depends on the concrete AD system in use. For example, some AD systems do not allow mutation of arrays (e.g. JAX (Bradbury et al., 2018)), while others do (e.g. Enzyme (Moses and Churavy, 2020)). Many AD systems allow for control flow, so that functions

with discontinuities as found in ESMs are differentiable by AD, even though they are not differentiable in a mathematical sense. Similarly, models with stochastic components, such as variational autoencoders (VAE), are also accessible for AD.

The defining feature of differentiable models is the efficient and automatic computation of gradients of functions of the model output with respect to (i) model parameters, (ii) initial conditions, or (iii) boundary conditions. Applying this paradigm to Earth System Models (ESMs) would enable gradient-based optimization of its parameters and the application of other methods that require information on gradients. For example, suppose for simplicity that the dynamics of an ESM may be represented, after discretization on an appropriate spatial grid, by an ordinary differential equation of the form,

$$\dot{\mathbf{x}} = f(\mathbf{x}, t; \mathbf{p}),$$

where $t$ denotes time, $\mathbf{x}(t)$ are the prognostic model variables that are stepped forward in time, and $\mathbf{p}$ are some parameters of the model. Optimization of the parameters $\mathbf{p}$ typically requires the minimization of a cost function $J(\hat{\mathbf{y}}, \mathbf{y})$, where $\hat{\mathbf{y}}$ is some function of a computed trajectory $\mathbf{x}(t; \mathbf{x}_0, \mathbf{p})$, with initial condition $\mathbf{x}_0 \equiv \mathbf{x}(0)$, and $\mathbf{y}$ is a known target value considered as

ground truth. A common choice of $J$ for regression tasks is the mean-squared error, $J(\hat{\mathbf{y}}, \mathbf{y}) = \frac{1}{N} \sum_{i=1}^{N} ||\hat{y}_i - y_i||^2$, where $N$ is the number of training examples. Gradient-based optimisation of $J$ with respect to the parameters $\mathbf{p}$ requires computing the derivative evaluations $\frac{\partial J}{\partial \mathbf{p}}(\hat{\mathbf{y}}, \mathbf{y})$, which in turn requires computing $\frac{\partial \hat{\mathbf{y}}}{\partial \mathbf{p}}(\mathbf{x}(t; \mathbf{x}_0, \mathbf{p}))$. Once the evaluations of the derivatives have been computed, the model parameters $\mathbf{p}$ can be updated iteratively in order to drive the predicted value $\hat{\mathbf{y}}$ towards the target value $\mathbf{y}$, thereby reducing the value of $J$. An identical procedure applies in the case of optimization with respect to initial

conditions or boundary conditions of the model.

Crucially, differentiable programming allows these derivatives to be computed for arbitrary choices of $\hat{\mathbf{y}}$. In the case of conventional ESM tuning, $\hat{\mathbf{y}}$ could be chosen to tune the model with respect to the global mean radiative balance, or global mean temperature, or both. A similar approach can be used to tune subgrid parameterizations, e.g. of convective processes in order to produce realistic distributions of cloud cover and precipitation. More generally, gradient-based optimization could be

used to train fully ML-based parameterizations of subgrid-scale processes, in which case gradients with respect to the network weights of an ANN are required.

Taken together, such approaches would directly move forward from the mostly applied manual and subjective parameter tuning to transparent, systematic, and objective parameter optimization. Moreover, automatic differentiability of ESMs would

provide an essential prerequisite for the integration of data-driven methods, such as ANNs, resulting in hybrid ESMs (Irrgang
et al., 2021).

## 4   Differentiable Models: Manual and Automatic Adjoints

Aside from AD, another approach to differentiable models is to manually derive and implement an adjoint model, usually from
a tangent linear model. This is especially common in GCMs that have been used for numerical weather prediction, as data
assimilation schemes such as 4D-Var (Rabier et al., 1998) also perform a gradient-based optimization to find initial states of
the model that agree with observations. This procedure takes a considerable amount of work. While such models can profit
from many of the advantages and possibilities of differentiable programming, this comes at the cost of missing flexibility and
customizability. Upon any change in the model, the adjoint has to be changed as well. Practitioners therefore need a very
good understanding of two largely separate code bases. In contrast, differentiable programming only needs manually defined
adjoints in a very small number of cases, mostly for more elementary operations such as Fourier transforms or the integration
of pre-existing code that is not directly accessible to AD. Differentiable models would also greatly simplify this process, as
already demonstrated with ANN-based emulators of GCMs (Hatfield et al., 2021). Another advantage of differentiable models
is that they can also automatically and efficiently compute second derivatives that can enable further optimization techniques.
In contrast, manually defining and maintaining a separate model for the Hessian is not realistically feasible.

Adjoint models of several ESM components have already been generated automatically with AD tools such as Transforms
of Algorithms in Fortran (TAF) (Giering and Kaminski, 1998) and Tapenade (Hascoët and Pascual, 2013). TAF is a library
that provides AD to generate code of adjoint models and has been successfully applied to GCMs like the MITgcm (Marotzke
et al., 1999) and PlaSIM (Lyu et al., 2018). In contrast to more modern AD systems, which work in the background without
the additionally generated code and structures for the derivative or adjoint exposed to the user, TAF directly translates and
generates code for an adjoint model and exposes it to the user. It comes with its own set of limitations that make it harder to
incorporate e.g. GPU use, or techniques and methods from ML and often needs user modifications to the generated adjoint
code. However, it has already led to many successful studies that show advances in parameter tuning (Lyu et al., 2018),
state estimation (Stammer et al., 2002) and uncertainty quantification (Loose and Heimbach, 2021). While these can be seen as
pioneering efforts for differentiable Earth system modeling, modern, more capable AD systems in combination with machinery
originally developed for ML tasks promise substantially greater benefits. Modern AD systems like the aforementioned JAX,
Zygote and Enzyme provide gradients in a more automaticated way, i.e. with less user interaction, and with greater generality
than AD systems like Tapenade while still also profiting from compiler optimisations and offering much easier interfacing
with ML workflows and potential GPU acceleration Innes et al. (2019). Each of these AD systems uses different approaches to
derive optimized gradient code and to assure manageable memory demand (see Sec. 6).

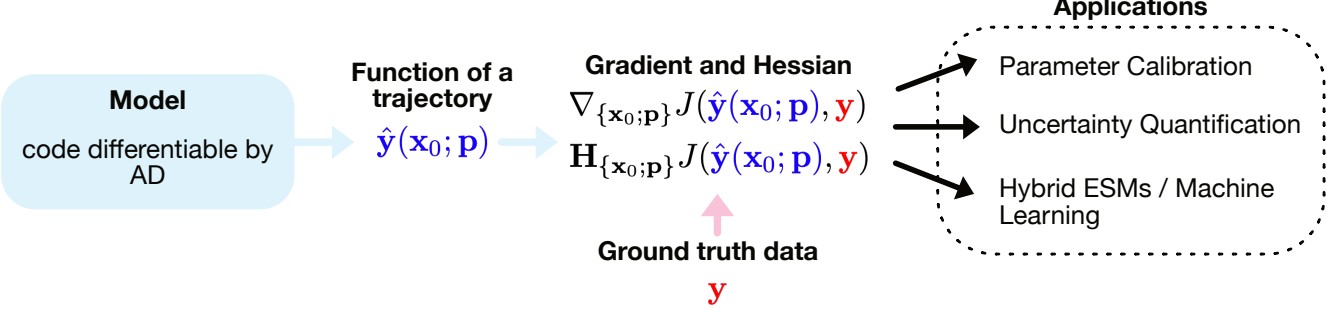

**Figure 1.** Differentiable programming enables gradient- and Hessian-based optimization of ESMs. Usually, gradients of a cost function $J$ that measures a distance between a desired function evaluation of a trajectory $\hat{y}$ and ground truth data $y$, e.g. from observations, are computed. The main benefits of this approach that we outline in this article are parameter calibration, uncertainty quantification and the integration of ML methods into Hybrid ESMs

## 5 Benefits of Differentiable ESMs

Fully automatically differentiable ESMs would enable the gradient-based optimization of all model parameters. Moreover, in the context of data assimilation, AD would strongly facilitate the search for optimal initial conditions for a model in question, given a set of incomplete observations. In the context of Earth system modeling, AD could lead to substantial advances mainly in three fields (Fig. 1): (a) Parameter tuning and parameterization, (b) probabilistic programming and uncertainty quantification, (c) integration of information from observational data via ML, leading to "Hybrid ESMs". Aside from these benefits for ESMs,

automatic differentiability would also offer significant benefits for data assimilation. The tangent-linear and adjoint model could be generated automatically, in contrast to the often manually derived adjoint models, as outlined in the previous section. Here, however, we will focus on the three aforementioned benefits for ESMs.

### 5.1 Parameter Tuning and Parameterization

Gradient-based optimization as facilitated by AD would allow for transparent, systematic, and objective calibration of ESMs

(see Sec. 2). This means that a scalar cost function of model trajectories and calibration data is minimized with respect to the ESM's parameters. The initial values of the parameters in this optimization would likely be based on expert judgment. Which parameters are tuned in such a way, and with respect to which (observational) data, is open to the practitioner but should be clearly and transparently documented. In principle, all parameters can be tuned, or a selection of individual parameters could be systematically tuned separately. Tsai et al. (2021) showed in a study based on an emulator of a hydrological land-surface

model that gradient-based calibration schemes that tune all parameters scale better with increasing data availability. They also show that even diagnostic variables, which are not directly calibrated, show better agreement with observational data with the gradient-based tuning. Additionally, extrapolation to areas from which no calibration data was used also performs better. A

fully differentiable ESM would likely enable these benefits without the need of an emulator, as demonstrated by promising results for the adjoint model of the relatively simple PlaSim model show (Lyu et al., 2018).

## 5.2 Probabilistic Programming and Uncertainty Quantification

Uncertainties in ESMs can stem either from the internal variability of the system (aleatoric uncertainties), or from our lack of knowledge of the modeled processes or data to calibrate them (epistemic uncertainty). In climate science, the latter is also referred to as model uncertainty, consisting of structural uncertainty and parameter uncertainty. Assessing these two classes of epistemic uncertainty is crucial in understanding the model itself and its limitations, but also increases reproducibility of studies conducted with these models (Volodina and Challenor, 2021). AD will mainly help to quantify and potentially reduce parameter uncertainty, whereas combining process-based ESMs with ML components and training the resulting hybrid ESM on observational data may help to address structural uncertainty as well.

Regarding parameter uncertainty, of particular interest are probability distributions of parameters of an ESM, given calibration data and hyperparameters; see e.g (Williamson et al., 2017) for a study computing parameter uncertainties of the NEMO ocean GCM. While there are computationally costly gradient-free methods to compute these uncertainties, promising methods such as Hamiltonian Markov Chains (Duane et al., 1987) need to compute gradients of probabilities, which is significantly easier with AD (Ge et al., 2018). Aside from that, when fitting a model to data via minimizing a cost function, the inverse Hessian, which can be computed for differentiable models, can be used to quantify to which accuracy states or parameters of the model are determined (Thacker, 1989). This approach is used to quantify uncertainties via Hessian Uncertainty Quantification (HUQ) (Kalmikov and Heimbach, 2014). Loose and Heimbach (2021) used HUQ with the adjoint model of the MITgcm and demonstrated how it can be used to determine uncertainties of parameters and initial conditions to uncover dominant sensitivity patterns and improve ocean observation systems. Petra et al. (2014) and Villa et al. (2021) developed a framework for Bayesian inverse problems using gradient and Hessian information that was already successfully applied to model the flow of ice sheets.

## 5.3 Hybrid ESMs

Gradient-based optimization is not limited to the intrinsic parameters of an ESM; it also allows for the integration of data-driven models. ANNs and other ML methods can be either used to accelerate ESMs by replacing computationally costly process-based model components by ML-based emulators, or to learn previously unresolved influences from data. It is also possible to combine ANNs with process-based physical equations of motion, e.g. via the Universal Differential Equation framework (Rackauckas et al., 2020), which allows for the integration and, more importantly, training of data-driven function approximators such as ANNs inside of differential equations. Usually trained via adjoint sensitivity analysis, this method also requires the model to be differentiable.

### 5.3.1 Emulators

Many physical processes occur on scales too small to be explicitly resolved in ESMs, for example the formation of individual clouds. To nevertheless obtain a closed description of the dynamics, parameterizations of the processes operating below the grid scale are necessary. While the training process of ANNs is computationally expensive, once trained, their execution is usually computationally much cheaper than integrating the physical model component that they emulate. Rasp et al. (2018) demonstrated successfully how an ANN-based emulator of a cloud-resolving model can replace a traditional subgrid parameterization in a coarser-resolution model. Similarly, Bolton and Zanna (2019); Guillaumin and Zanna (2021) demonstrated an ANN-based subgrid parameterization of eddy momentum forcings in ocean models and Jouvet et al. (2022) introduced a hybrid ice sheet model in which the ice flow is a CNN-based emulator that accelerates the model by several orders of magnitude. While emulators would usually be trained offline first (i.e., outside of the ESM), a differentiable ESM would enable fine-tuning of the ANN inside the complete ESM.

### 5.3.2 Modeling Unresolved Influences

A growing number of processes in ESMs are not based on fundamentally known primitive equations of motion, such as the Navier-Stokes equation of fluid dynamics for the atmosphere and oceans. For example, vegetation models are typically not primarily based on primitive physical equations of the underlying processes, but rather on effective empirical relationships and ecological paradigms. For suitable applications, many ESM components, such as those describing land-surface and vegetation processes, ice sheets, or the carbon cycle, or subgrid-scale process in the ocean and atmosphere, could be replaced or augmented with data-driven ANN models. Even components that are based on known primitive equations of motion have to be discretized to finite spatial grids in practice, so that they can be integrated numerically. The resulting parameterizations will necessarily introduce errors that can be attenuated by suitable data-driven and especially ML methods. For example, Um et al. (2020) demonstrated that a hybrid approach can reduce the numerical errors of a coarsely resolved fluid dynamics model by showing that a fully differentiable hybrid model on a coarse grid performs best in learning the dynamics of a high-resolution model. With a similar approach, Kochkov et al. (2021) showed that a hybrid model of fluid dynamics can result in a 40- to 80-fold computational speedup while remaining stable and generalizing well. Zanna and Bolton (2021) also propose physics-aware deep learning to model unresolved turbulent processes in ocean models. Universal Differential Equations (UDEs) can be such a physics-aware form of ML, as they can combine primitive physical equations directly with ANNs to minimize model errors effectively. de Bézenac et al. (2019) developed a hybrid model to forecast high-resolution sea surface temperatures by using the output of an ANN as input for a differentiable advection-diffusion model, outperforming both coarse process-based models and purely data-driven approaches.

Ideally, a hybrid ESM could combine both of these approaches (emulation and modeling of unresolved processes) and perform its final parameter optimization for both the physically motivated parameters and the ANN parameters at once, in the full hybrid model. Differentiable programming would enable such a procedure. Differentiable ESMs are thus prime candidates for strongly coupled neural ESMs in the terminology of Irrgang et al. (2021).

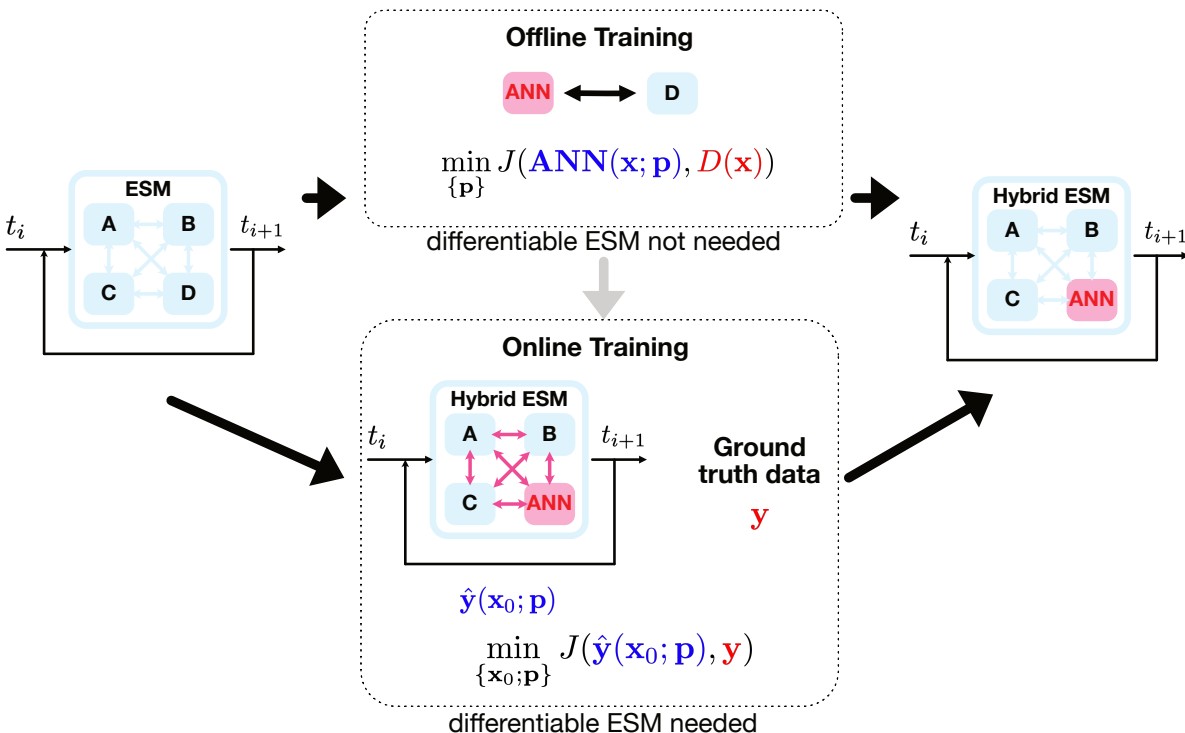

**Figure 2.** When replacing or augmenting parts of an ESM with ML methods such as ANNs, one has to train the ANN by minimizing a cost function $J$ that measures a distance between the output of the ML method and the ground truth data. There are two fundamentally different ways to set up this training: (i) offline training, in which the component is trained outside of the ESM, e.g. to emulate another component (here called "D"), and (ii) online training, in which gradients are taken through the operations of the complete model. Online training requires a differentiable ESM and leads potentially to more stable solutions of the resulting Hybrid ESM. Both approaches can be combined by pre-training an ML method offline before switching to an online training scheme as indicated by the grey arrow; the latter approach would be computationally less expensive.

Essentially, ESMs are algorithms that integrate discretized versions of differential equations describing the dynamics of processes in the Earth system. As such, a comprehensive, hybrid, differentiable ESM could constitute a UDE (Chen et al., 2018; Rackauckas et al., 2020). The gradients for such models are usually computed with adjoint sensitivity analysis that in turn also relies on AD systems (see Sec. 6 for challenges of computing these gradients). However, individual subcomponents such as emulators might also be pre-trained offline, outside of the differential equation and therefore without adjoint-based 270     methods, similar e.g. to the subgrid parameterization models reported in Rasp et al. (2018).

### 5.4  Online vs Offline Training of Hybrid ESMs

Existing applications of ML methods to subgrid parameterizations in ESMs generally follow a three-step procedure (Rasp, 2020):

1. Training data is generated from a reference model, for example a model of some subgrid-scale process that would be too costly to incorporate explicitly in the full ESM.

2. An ML model is trained to emulate the reference model or some part of it.

3. The trained ML component is integrated into the full ESM, resulting in a hybrid ESM.

Steps 1 and 2 constitute a standard supervised ML task. Following the terminology of Rasp (2020), we say that the ML model is trained in offline mode, that is, independently of any simulation of the full ESM. ML models which perform well offline can nonetheless lead to instabilities and biases when coupled to the ESM in online mode, that is, during ESM simulations. One possible explanation for the instability of models which are trained in offline mode is the effect of accumulating errors in the learned model when it is coupled in online mode, which can lead to an input distribution that deviates from the distribution experienced by the ML model during training (Um et al., 2020). When an ML model is not only supposed to be run in online mode, but also be trained online, the complete ESM needs to be differentiable because the cost function used in training does not only depend on the ML model, but on all parts of the ESM.

Recent work demonstrates the advantages of training the ML component in online mode using differentiable programming techniques. Um et al. (2020) show that adding a solver-in-the-loop, that is, training the ML component in online mode by taking gradients through the operations of the numerical solver, leads to stable and accurate solutions of PDEs augmented with ANNs. Similarly, Frezat et al. (2022) learn a stable and accurate subgrid closure for 2D quasi-geostrophic turbulence by training an ANN component in online mode, or as they call it, training the model end-to-end. Frezat et al. (2022) distinguish between a priori learning, in which the ML component is optimized on instantaneous model outputs (offline mode), and a posteriori learning, in which the ML component is optimized on entire solution trajectories (online mode, see Fig. 2). Both Um et al. (2020) and Frezat et al. (2022) find that models trained in offline mode lead to unstable simulations, underscoring the necessity of differentiable programming for hybrid Earth system modeling. Kochkov et al. (2021) also report stable solutions of their online-trained hybrid fluid dynamics model, which generalizes well to unseen forcing and Reynolds numbers.

An additional benefit of training a hybrid ESM in online mode is the ability to optimize not only with respect to specific processes, but also with respect to the overall model climate. An ML parameterization trained in offline mode will typically be trained to emulate the outputs of an existing process-based parameterization for a plausible range of inputs. However, even for models which perform very well offline, it is not known if they will produce a realistic climate until they are coupled to the ESM after training. In contrast, an equivalent ML parameterization trained in online mode can be optimized not only with respect to the outputs of the parameterization itself, but also to reproduce a realistic overall climate.

In a typical scenario for hybrid ESMs, e.g. an ANN-based subgrid parameterization, online learning can also lead to more stable and accurate solutions, as showcased by studies in fluid dynamics (Frezat et al., 2022; Um et al., 2020; Kochkov et al., 2021). However, training a subcomponent of an ESM online is computationally more expensive than training it offline. Therefore, it seems reasonable to combine both approaches and start pre-training offline before switching to a potentially necessary online training scheme.

# 6    Challenges of Differentiable ESMs

While the benefits of differentiable ESMs are extremely promising, they come at a cost. Every AD system has certain limitations and there might not even exist a capable AD system in the programming language in which an existing ESM is written. Many state-of-the-art AD systems have been designed with ML workflows in mind, which usually consist of pure functions with only limited support for array mutation and in-place updates (Innes et al., 2019; Bradbury et al., 2018). Projects like Enzyme (Moses and Churavy, 2020) promise to change that. However, even with more capable AD systems, converting existing ESMs to be differentiable is a challenging task: it potentially requires the translation of the model code to another programming language and at least a major revision of the code to work with one of the suitable frameworks or AD tools. Rewriting an ESM in a differentiable manner has the potential co-benefit that such a rewrite can also incorporate other modern programming techniques such as GPU usage and the use of low-precision computing, both resulting in potentially huge performance gains (Häfner et al., 2021; Wang et al., 2021; Klöwer et al., 2022). In particular GPU acceleration has the potential to make ESMs faster and more efficient as e.g. demonstrated by the JAX-based Veros ocean GCM (Häfner et al., 2021). Given that ESMs are very complex models, tracking every single elementary operation in a tape by the AD might induce unfeasible overheads as e.g. remarked by (Farrell et al., 2013). dolfin-adjoint for FEM models solves that by differentiating on a higher abstraction level (Farrell et al., 2013; Mitusch et al., 2019). JAX models perform most operations as vector and tensor operations. (Häfner et al., 2018) deliver a good account of this vectorization process during the translation of their Veros model. Again, this process also has the potential co-benefit of GPU acceleration. Enzyme.jl works on the level of the LLVM IR which enables highly optimized gradient code. It will attempt to recompute most values in the reverse pass per default and cache (tape) only what is necessary (Moses and Churavy, 2020). Zygote.jl uses a static single assignments form that is more efficient and also recomputes values in the reverse mode instead of storing everything (Innes et al., 2019).

Memory demand is a fundamental challenge when computing gradients of functions of trajectories of ESMs over many timesteps, saving all intermediate steps needed to compute the gradient requires a prohibitively large amount of RAM. Therefore, checkpointing schemes have to balance memory usage with recomputing intermediate steps. There are different schemes available to do so, such as periodic checkpointing or Revolve that try to optimize this balance (Dauvergne and Hascoët, 2006; Griewank and Walther, 2000). For example, for their differentiable finite volume PDE solver (Kochkov et al., 2021) remark that every time step is checkpointed.

ESMs utilize different discretization techniques and solvers: (pseudo-)spectral, finite-volume, finite-element, and other approaches can be used in the different components of ESMs. Differentiable modeling is possible for all of these approaches in principle. While this is relatively straightforward for spectral models, it has also been demonstrated for finite-volume and finite-element solvers (Souhar et al., 2007; Kochkov et al., 2021; Farrell et al., 2013). In particular, dolfin-adjoint (Mitusch et al., 2019) for the popular FEniCS and Firedrake FEM libraries (Logg et al., 2012; Rathgeber et al., 2016) are available and easily applicable for existing FEM models. (Kochkov et al., 2021) demonstrate in their work and related software differentiable CFD PDE solvers that can use both finite volume and pseudo-spectral approaches using JAX and with GPU/TPU acceleration.

Solvers can also make use of AD during their forward computation, e.g. when solving the involved nonlinear equation systems. Some solvers also have to make use of slope or flux limiters to eliminate spurious oscillations close to discontinuities of the solution (see e.g. (Berger et al.) for an overview). Ideally those slope limiters should also be differentiable (Michalak and Ollivier-Gooch, 2006), however as AD can differentiate through control flow, limiters with continuous but not differentiable limiter functions might also work.

Aside from technical challenges, a more fundamental problem to address is the chaotic nature of the processes represented in ESMs. Nearby trajectories quickly diverge from each other, which makes optimization based on gradients of functions of trajectories error-prone if the practitioner is not aware of this. Often, gradients computed both from AD or iterative methods and adjoint sensitivity analysis are orders of magnitude too large because of ill-conditioned Jacobians and the resulting exponential error accumulation; see (Metz et al., 2021; Wang et al., 2014) for details. This is especially problematic when the recurrent
Jacobian of the system exhibits large eigenvalues (Metz et al., 2021). Luckily, there are some approaches to reduce this problem. For example, least-squares shadowing methods can compute long-term averages of gradients of ergodic dynamical systems (Wang et al., 2014; Ni and Wang, 2017). Such shadowing methods have already been explored for fluid simulations (Blonigan et al., 2017). Alternatively, the sensitivity and response can be computed using a Markov-chain representation of the dynamical system (Gutiérrez and Lucarini, 2020). The problem can also be addressed using the regular gradients, but with an iterative
training scheme, starting from short trajectories (Gelbrecht et al., 2021). In addition to the chaotic nature of ESMs, they also usually constitute so-called stiff differential equation problems, caused by the difference in time scales of the different modeled processes. Stiff differential equations can lead to additional errors when using reverse-mode AD or adjoint sensitivity analysis. These errors can be mitigated by rescaling the optimization function and choosing appropriate algorithms for the sensitivity analysis as outlined by Kim et al. (2021).

An additional challenge for differentiable ESMs is the inclusion of physical priors and conservation laws. While the parameters of a differentiable ESM may generally be varied freely during gradient-based optimization, it is nonetheless desirable that they should be constrained to values which lead to physically consistent model trajectories. This challenge is particularly acute for hybrid ESMs, in which some physical processes may be represented by ML model components with many optimizable parameters. Enforcing physical constraints is an essential step towards ensuring that hybrid ESMs which are tuned to present-day climate and the historical record will nonetheless generalize well to unseen future climates.

A number of approaches have been proposed to combine physical laws with ML. Physics-informed neural networks (Raissi et al., 2019) penalize physically inconsistent solutions during training, the penalty acting as a regularizer which favors, but does not guarantee, physically consistent choices of the model weights. Beucler et al. (2019, 2021) enforce conservation of energy in an ML-based convective parameterization by directly constraining the neural network architecture, thereby guaranteeing
that the constraints are satisfied even for inputs not seen during training. Another architecture-constrained approach involves enforcing transformation invariances and symmetries in the ANN based on known physical laws (Frezat et al., 2022). In some cases it is possible to ensure that physical constraints are satisfied by carefully formulating the interaction between the ML- and process-based parts of the hybrid ESM. For example, Yuval et al. (2021) ensure conservation of energy and water in an ANN-based subgrid parameterization by predicting fluxes rather than tendencies of the prognostic model variables.

# 7 Conclusions

The ever-increasing availability of data, recent advances in AD systems, optimization methods and data-driven modeling from ML, create the opportunity to develop a new generation of ESMs that are automatically differentiable. With such models, long-standing challenges like systematic calibration, comprehensive sensitivity analyses and uncertainty quantification can be tackled and new ground can be broken with the incorporation of ML methods into the process-based core of ESMs.

Ideally, every single ESM component, including couplers, would need to be differentiable, which of course takes a considerable amount of work to realize. Differentiable programming requires different programming languages and styles than have been common practice in ESMs. Automated code translation might assist this process in the future, as ML-based tools like ChatGPT (OpenAI, 2022) showed great promise even in complex programming tasks, can understand Fortran code and can be instructed to use libraries like JAX in their translation. Despite this, we are fully aware that translating model code is a very tedious business. Nevertheless, future model development should take differentiable programming into account to get the tremendous benefits that we outlined in this article.

If almost all components of an ESM are differentiable, but one component is not, one might be still be able to achieve a fully differentiable model through implicit differentiation, as recent advances also work towards automating implicit differentiation (Blondel et al., 2021). While most of the research discussed so far focuses on ocean and atmosphere GCMs, differentiable programming techniques certainly have huge potential for other ESM components like biogeochemistry, terrestrial vegetation or ice sheet models that already incorporate more empirical relationships. Differentiable models can leverage the available data better in these cases, both improving calibration and the incorporation of ML subcomponents to model previously unresolved influences.

For the calibration of ESMs, differentiable programming enables not only a gradient-based optimization of all parameters together, but also more carefully chosen procedures where expert knowledge is combined with the optimization of individual parameters. Similarly, the cost function that is used in the tuning process can be easily varied and experimented with. The ability to objectively optimize all parameters for a differentiable ESM does of course not imply that all parameters should be optimized. Rather, differentiable ESMs allow documenting which parameters are calibrated to best reproduce a given feature of the Earth system in a transparent manner.

Where previous studies had to use emulators of ESMs to showcase the potential of differentiable models, fully differentiable ESMs can harness this potential while maintaining the process-based core of these models. Differentiable ESMs also enable this process-based core to be supplemented with ML methods more easily. Deep learning has shown enormous potential, e.g. for subgrid parameterization, to attenuate structural deficiencies and to speed up individual, slow components by replacing them with ML-based emulators. Differentiable ESMs make this process easier. The possibility of online training of ML models within ESMs promises to lead to more accurate and stable solutions of the combined hybrid ESM.

Aside from this, differentiable ESMs also enable further studies on the sensitivity and stability of the Earth's climate, which previously had to rely on gradient-free methods. For example, algorithms to construct response operators to further study how fluctuations, natural variability, and response to perturbations relate to each other (Ruelle, 1998) can be implemented with a

differentiable model. Advances in this direction would greatly improve our understanding of climate sensitivity and climate change (Lucarini et al., 2017).

Differentiable ESMs are a crucial next step toward improved understanding of the Earth's climate system, as they would be able to fully leverage increasing availability of high-quality observational data and and to naturally incorporate techniques from ML to combine process understanding with data-driven learning.

*Author contributions.* MG led the preparation of the manuscript. All authors discussed the outline and contributed to writing the manuscript.

*Competing interests.* The authors declare that they have no competing interests.

*Code and data availability.* No code or data sets where used for this article.

*Acknowledgements.* This work received funding from the Volkswagen Foundation. NB acknowledges further funding from the European Union's Horizon 2020 research and innovation programme under grant agreement No. 820970 and under the Marie Sklodowska-Curie grant agreement No. 956170, as well as from the Federal Ministry of Education and Research under grant No. 01LS2001A. This is TiPES contribution #X.

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
