# Peer review of "Differentiable Programming for Earth System Modeling"

_EGUsphere, 2022_

## Referee Comment (RC1)

**Review of egusphere-2022-875: *Differentiable Programming for Earth System Modeling**

By Maximilian Gelbrecht, Alistair White, Sebastian Bathiany, Niklas Boers

Reviewer: Sam Hatfield, ECMWF

26 Oct 2022

Verdict: **Accept as is**

In this review the authors present a strong, comprehensive, yet succinct argument for developing differentiable Earth system models. The benefits of differentiability are discussed in the contexts of model parameter tuning, uncertainty quantification, and "hybrid" Earth system models, in which tools from machine learning are interfaced with a traditional model online. Some difficulties that may be encountered in practice are also outlined. Overall the manuscript serves as an excellent introduction to the ideas behind differentiability and would make for a useful reference in the literature as this idea gains traction. I would be happy to accept the manuscript as-is. Reading the manuscript, I did have some comments, but these are just left for the authors' consideration and are not actionable:

- Speaking as someone who works with an operational weather prediction system that relies on a variational data assimilation algorithm, 4D-Var, I am most interested in the idea of differentiability in the context of tangent-linear and adjoint model generation. An inherently differentiable model could in principle obviate the need to hand-code tangent-linear and adjoint versions of the nonlinear forecasting model, and the authors briefly discuss this in Section 4. This was also the motivation behind my paper on neural network-based tangent-linear and adjoint models which the authors kindly cited. This idea could be expanded upon as a fourth benefit for differentiability in addition to those in Section 5. However I think in that case it would be wise to consult someone with direct experience with data assimilation development at ECMWF (Alan Geer would be the best person). There have been efforts to apply established automatic differentiation tools to the ECMWF model source code in the past, but as far as I know none of these amounted to anything useful, and ECMWF still relies on completely handwritten tangent-linear and adjoint code.
- Are the authors aware of the work of Stephan Hoyer, Peter Norgaard, Dmitrii Kochkov, and others at Google? If I understand both their work and this manuscript correctly, they are essentially attempting to do just what the authors propose. They have already published on direct numerical simulation of fluid flow, but they are currently developing a differentiable spectral general circulation model in JAX-CFD with learned physics. I don't believe there is anything citable on this work yet, but it's worth keeping an eye on it.

---

## Author Response (AR1)

Maximilian Gelbrecht
Earth System Modelling Group
TUM School of Engineering and Design
Technical University Munich
Lise-Meitner-Straße 9, 85521 Ottobrunn, Germany
email: maximilian.gelbrecht@tum.de

[Figure]

Munich, December 19, 2022

Dear Editors of Geoscientific Model Development,

Please find a point-by-point response to the comments by both referees below. The comments of the referees are shown in blue font.

**Review 1**   We thank the Reviewer for the positive assessment of our article. With regards to the inclusion of benefits for Data Assimilation (DA) and NWP, we can fully understand the reviewer's suggestion to optionally include it. In fact, we also discussed this within our author group before. Ultimately, in this article, we want to focus on Earth System Modelling and the benefits and challenges of differentiable programming therein. In the revised version of the article we added a paragraph to the benefits section, in which we also mention the potential and benefits for DA and NWP, similar to the already included references in the adjoint section. However, we would welcome the opportunity to extend this into a possible follow-up article, for which we would need the expertise of the reviewer or other experts like Alan Geer, as DA and NWP are not within our core areas of expertise.

We are aware of the ongoing research of the team at Google that the reviewer mentioned. To our knowledge they haven't published it yet. The involved scientists like Stephan Hoyer and Dimitri Kochkov did however publish noteworthy papers on scientific machine learning and how to integrate knowledge into machine learning methods before. We included their paper on machine learning accelerated fluid dynamics in the revised manuscript.

**Review 2**

- Many relevant works are not cited. There are several PDE software packages that employ differentiable programming: FEniCS, Firedrake, and devito, to name a few in chronological order. While not originally built to use differentiable programming, it is also possible in deal.II using dual numbers. There are yet more software packages built on these tool kits for modeling individual components of the earth system, for example Gusto, Thetis, icepack, VarGlaS. Granted, no one has built, say, a coupled atmosphere-ocean GCM using these tools, but they're worth mentioning nonetheless The biggest omission regarding differential programming for PDE solvers is Farrell et al (2013), Automated derivation of the adjoint of high-level transient finite element programs. This paper won the SIAM Wilkinson Prize for Numerical Software in 2015.

- We thank the reviewer for this comment. So far in our article we didn't go into detail on the different discretisation techniques that ESMs use. As far as we know, all of the projects that the reviewer lists here are concerning FEM methods. FEM methods are just one possible discretisation technique: (pseudo-)spectral, finite differences, finite volume and other forms of discretisation also all play a role for ESMs. We consider it therefore outside of the scope

of the article to go into a lot of detail on FEM solvers. Even regular ODE solvers can profit from determining the Jacobian more efficiently and precise via AD. However, in our revised article we added a paragraph on discretisation techniques in general. In this paragraph we mention that one can realize differentiable ESMs independent of the chosen discretisation method, as e.g. demonstrated by the Farrell paper that the reviewer suggested, which shows that differentiable programming can also be applied to FEM models. Additionally, in our revised article we also include additional references showcasing the prior research on combining ML techniques with ice-sheet models. We also added further references e.g. to research on combining fluid dynamics with machine learning.

- 32, "Modern AD systems are able to differentiate most typical operations that appear in ESMs": What about flux or slope limiters? Do you believe in discretize then optimize, or the other way around?

- In general, slope limiters can also be part of differentiable ESMs. See e.g. Fikl et al (arXiv:2209.03270v1) for an adjoint based optimization including slope limiters. There are slope limiters that are differentiable in a mathematical sense, and some that are not. AD can differentiate through control flow, so that even slope limiters like minmod that are not differentiable in a mathematical sense might be possible to use. However, this will depend on the practical implementation of the ESM component that includes the slope limiter, its solvers, discretisation and the way the gradient is computed. We are not aware of research investigating this in detail and in the revised manuscript we therefore added comments on slope limiters in the Challenges of Differentiable ESMs section.

- 40, "Third, additional information from observations can be integrated into ESMs with Machine Learning (ML) models." I'd say that ML tools enable you to construct very complex statistical models and train them with the data you have, but ML as such does not somehow enable you to integrate more information from this data into process-based physical models of the earth system than you could with a more old-school statistical system identification or parameter estimation viewpoint. This is classic information theory, see Kullback's 1958 book.

- We thank the reviewer for this comment, but it is not quite clear to us to which old-school statistical system identification or parameter estimation methods they refer. Machine learning methods like artificial neural networks are also not really new. They built upon statistics and optimisation theory like many other methods. ANNs however do provide extremely flexible universal function approximators that, through their very high capacity, are able to model more complex behaviour than many other methods. In our article we also cite various papers, e.g. the work from Um et al., Yuval et al., and Rasp et al., which showcase how ANNs can be used to improve a more traditional subgrid parametrisation. The point we were trying to make here is that, once a process-based ESM (component) is formulated such that it is automatically differentiable, it will also be much easier to seamlessly combine it with ML components; in addition, optimizing both the parameters of the process-based component and the parameters of the ML part will only be possible if both are automatically differentiable.

- 91-94, "Artificial neural networks (ANNs) can be seen as a subset of these models, but differentiable programming goes far beyond these building blocks": A lot of the wording here is conflating what problem you're trying to solve with how you're trying to solve it. Fitting the parameters of a model, whether it's an ANN or process-based physics model, is the answer to the "what" question. There are many ways you could solve this fitting problem. You could use derivative-free optimization methods – it's not a very good idea,

but you could do it. Using gradient-based optimization methods is the answer to a "how" question, and using AD to compute the gradient as opposed to deriving it on pen and paper (which you can still do for some PDE models) is a subset of that "how" question. The fact that you can differentiate through control flow or user-defined types is definitely a compelling reason to use AD. You do address this and quite well in section 4, but it's really important to make the distinction clear.

- We thank the reviewer for their careful review of this section; indeed we should have made this clearer. In the revised manuscript, we revised this section. The first two paragraph are concerning the "how", and the last two paragraphs the "what" from the perspective of an Earth system modeler.

- 170: I think it's worth making a bigger deal out of the fact that you can get the second derivative so easily with AD. It's often painful but still possible to manually derive a first-order adjoint model, but going to second order by hand is really atrocious.

- We agree with the reviewer that computing second derivatives can have considerable benefits in theory, and we do mention this in a number of places in the manuscript, e.g. in the overview figure. In theory, this can indeed have considerable benefits. However, this also comes at a cost. In particular, computing the Hessian can consume too much memory to be worth considering. Often, methods rather try to estimate a Hessian-vector-product in ways that don't actually need second derivatives at all. That being said, if more models and tools are able to compute second derivatives easily, it is possible that more algorithms will be developed that might avoid the huge memory cost of the full Hessian and actually use proper second derivatives. Therefore we added a comment on Hessians to the manual vs automatic adjoint section.

- 175: Here it's worth citing some of Noemi Petra's work, including here paper on stochastic Newton MCMC as well as her more recent work on hIPPYlib.

  We thank the reviewer for pointing us to this work and we added it to the revised manuscript.

On behalf of the authors,

sincerely,

Maximilian Gelbrecht

---

## Author Response (AR2)

Maximilian Gelbrecht
Earth System Modelling Group
TUM School of Engineering and Design
Technical University Munich
Lise-Meitner-Straße 9, 85521 Ottobrunn, Germany
email: maximilian.gelbrecht@tum.de

[Figure]

Munich, April 12, 2023

Dear Dr Ham,

Thank you very much for your in-depth look at our paper. We agree that we were brushing over some potential concerns and questions that geoscientific model developers might have a bit too quickly in our previous revision. Therefore we have revised the manuscript further.

In our revised article we are much clearer on past advances that have been made with tools such as TAF and Tapenade, as well as the FEM models that can use dolphin-adjoint for deriving differentiable models. Certainly these approaches can contribute massively to the potential benefits of differentiable ESMs. We also agree with the editor that for a comprehensive ESM to be differentiable it might not be feasible for AD to tape each elementary operation. This is why approaches like the one taken by dolphin-adjoint are extremely valuable. Alternatives to that depend on the AD system in question.

With JAX, we see a strong vectorization of the code together with checkpointing schemes as an alternative to that. For their differentiable FVM PDE solver Kochkov et al remark that every time step is checkpointed. As the editor remarked, this is also the approach taken by most ML libraries. Adapting an existing model e.g. to the JAX framework wouldn't even be possible without vectorization. Häffner et al deliver a detailed account of that process in their GMD paper on the JAX-based Veros ocean model (albeit their full model is not differentiable yet). Enzyme works on the level of the LLVM IR which enables highly optimized gradient code. It will attempt to recompute most values in the reverse pass per default and cache (tape) only what is necessary (Moses, 2020). Zygote uses static single assignments form that is more memory efficient than "normal" tapes / Wengert lists and also recomputes (checkpoints) values in the reverse mode (Innes, 2019).

In the revised version we also reference more directly the recent success of differentiable CFD solvers of Kochkov et al that employ a FVM and pseudo-spectral approach in their JAX-CFD library. Their study using this framework clearly showcases the enormous potential differentiable models can have as they show an enormous speed-up in their simulations. All of this comes with the additional co-benefit of GPU acceleration as well. Ongoing projects such as. the DJ4Earth and CliMA project also work on a differentiable version of the FVM ocean model Oceananigans. We also remark that the current generation of AD tools needs much less user intervention as the editor is also absolutely correct in remarking the massive manual intervention needed for adjoint models like MITgcm.

However, it is also important to stress that not all ESM components are FEM models and not all ESM components are AO-GCMs. Let's consider vegetation models for example. To

our knowledge currently there is no differentiable vegetation model. Therefore, other tools are needed for those components and these can be vectorized models using frameworks such as JAX or Julia's SciML ecosystem.This is something that we are indeed also working on, as well as differentiable pseudo-spectral atmosphere models.

Indeed we were also a bit too quick to reject the challenges imposed by flow control. AD can without major problems differentiate many functions that are continuous but not differentiable, as e.g. the ReLU function. Differentiating through discontinuous functions does indeed involve the problems the Editor is mentioning. We are fully aware that achieving differentiable comprehensive ESMs is a very challenging task that also involves further research and a lot of tedious model translations. This is a perspective article, its purpose is also to offer a perspective, a personal viewpoint and look forward. We were motivated by the recent success of projects such as those of Kochkov et al Google Research group and Um, Holl and Thurey et al. at with their differentiable PDE solvers based on frameworks such as JAX, the ongoing work on the fully Julia-based CliMA model and our own ongoing work on a differentiable pseudo-spectral atmosphere model to propose this perspective. We believe that future model development should take differentiable programming into account to get the tremendous benefits that we outlined in this article.

Section 3 that mostly explains basic principles of AD has already been revised in response to Reviewer 2s suggestions, we added to Section 4 and 6 to explain in more detail what practitioners have to do and how this compares to past efforts with tools such as TAF and Tapenade.

We also checked our manuscript again carefully for the issues raised by Reviewer 2 and have addressed them point-by-point as follows:

- We added, compared to the original manuscript, many references, especially those referring to FEM such as dolphin-adjoint for the FEniCs and Firedrake frameworks, and past efforts for differential ESMs with tools such as TAF and Tapenade. We also extended the references to differentiable PDE solvers for CFD such as JAX-CFD of Kochkov et al and PhiFlow of Holl et, and Um et al. We hope that this reflects past and ongoing work on differentiable models more broadly

- We refer to slope limiters in the Challenges for differentiable ESMs section

- Additional references to differentable PDE solvers and studies that showcase the enormous potential of ANNs to accelerate and complement them have been added. In Section 5.3 we explain several studies that show the enormous potential for hybrid approaches that combine physics-based models with ANNs, e.g. an ANN subgrid scale parameterization. We believe that these approach would not be possible with more "old-school statistical methods",

- We revised Section 3 already in our last revision to make it much clearer what the "how" and the "what" is from the perspective of an Earth system modeler. In this revision we extended upon that in Section 4 and Section 6, to make it more clear what practical steps would have to be undertaken for a differentiable ESM

- We agree with the reviewer that computing second derivatives can have considerable benefits. While there are also computationally very intensive to compute, we already added more references to this possibility in the last revision

- We added a reference Noemi Petra's work to our manuscript as the Reviewer recommended

On behalf of the authors,

sincerely,

Maximilian Gelbrecht